# Unveiling the Impact of Human Herpesviruses-Associated on CNS Infections: An Observational Study

**DOI:** 10.3390/v16091437

**Published:** 2024-09-09

**Authors:** Caio Cesar L. B. Barrionuevo, Pedro P. A. Baptista, Ewerton F. da Silva, Bernardo M. da Silva, Cássia da L. Goulart, Sabrina A. de Melo, Valderjane A. da Silva, Lara Laycia A. de Souza, Rossicleia L. Monte, Fernando F. Almeida-Val, Pablo Vinícius S. Feitoza, Michele de S. Bastos

**Affiliations:** 1Programa de Pós Graduação em Medicina Tropical, Universidade do Estado do Amazonas, Manaus 69055-038, Brazil; caioleivabastos@gmail.com (C.C.L.B.B.); bernardo.mpesq88@gmail.com (B.M.d.S.); cassiadaluzgoulart@gmail.com (C.d.L.G.); laralaycia@gmail.com (L.L.A.d.S.); 2Faculdade de Educação Física e Fisioterapia, Universidade Federal do Amazonas, Manaus 69060-001, Brazil; pedropoa@ufam.edu.br; 3Gerência de Bacteriologia, Fundação de Medicina Tropical Doutor Heitor Vieira Dourado, Manaus 69040-000, Brazil; ewertondsf1997@gmail.com (E.F.d.S.); sabrinaamello65@gmail.com (S.A.d.M.); rossi@fmt.am.gov.br (R.L.M.); michelebastos99@gmail.com (M.d.S.B.); 4Programa de Pós-Graduação em Ciências da Saúde (PPGCIS), Manaus 69020-180, Brazil; valderjane@gmail.com; 5Departamento de Clínica Cirúrgica, Faculdade de Medicina, Universidade Federal do Amazonas, Manaus 69020-170, Brazil; pvsfeitoza@ufam.edu.br

**Keywords:** Human Herpesviruses, CNS infections, cerebrospinal fluid, HIV/AIDS, CSF analysis

## Abstract

Human Herpesviruses (HHVs) play a significant role in neurological diseases such as encephalitis and meningitis, adding significant morbidity. This study aims to retrospectively analyze the effect of HHVs on patients with neurological symptoms, focusing on the *Herpesviridae* family’s contributions to central nervous system (CNS) infections. Methods: This retrospective cohort study included 895 patients suspected of viral CNS infections, utilizing molecular diagnosis via qPCR to identify HHVs in cerebrospinal fluid (CSF) samples. This was conducted at a reference tertiary care hospital for infectious diseases in the western Brazilian Amazon from January 2015 to December 2022, focusing on the *Herpesviridae* family’s clinical repercussions and of Cytomegalovirus in CNS infections. Results: The findings revealed that 7.5% of the analyzed samples tested positive for HHVs, with Human *Cytomegalovirus* (HCMV) and *Epstein–Barr* Virus (EBV) being the most prevalent. A significant association was found between HHVs and neurological diseases such as encephalitis and meningitis, especially among people living with HIV/AIDS (PLWHA), highlighting the opportunistic nature of these viruses. The study underscores the critical role of CSF analysis in diagnosing CNS infections and the complexity of managing these infections in HIV patients due to their immunocompromised status. Conclusions: The results emphasize the need for comprehensive diagnostic approaches and tailored treatment strategies for CNS infections in immunocompromised individuals. The study calls for ongoing research and advancements in clinical practice to improve patient outcomes facing CNS infections, particularly those caused by HHVs.

## 1. Introduction

Human Herpesviruses (HHVs) are a significant group within the *Herpesviridae* family, characterized by their spherical virion structure, including a nucleus, capsid, tegument, and envelope. These viruses possess linear double-stranded DNA genomes ranging from 125 to 241 kb, encompassing 70 to 170 genes. They are classified into three subfamilies: *Alphaherpesvirinae*, *Betaherpesvirinae*, and *Gammaherpesvirinae*, each containing notable pathogens such as HSV-1, HSV-2, and *Varicella-Zoster* Virus (VZV) in *Alphaherpesvirinae*; Epstein–Barr Virus (EBV) in *Gammaherpesvirinae*; and Human *Cytomegalovirus* (HCMV) in *Betaherpesvirinae* [1].

HHVs are prevalent worldwide, often acquired during childhood or through sexual contact [2,3]. These viruses can remain latent within the host and be reactivated by various factors, including immunosuppression and stress [4]. Globally, about 3.7 billion individuals carry HSV-1 and 417 million have HSV-2, with the most prevalent cases found in areas with scarce resources. Vaccine efficacy has been shown in real-world settings, i.e., for VZV, including in immunocompromised individuals; nonetheless, effective prophylactic EBV vaccines have faced major obstacles [5,6,7].

These five HHVs can infect the CNS, leading to severe conditions such as encephalitis and meningitis [8], though they do not always result in neurological syndromes. They also contribute to other serious health issues, including autoimmune diseases and neurodegenerative disorders such as Alzheimer’s, Parkinson’s, and multiple sclerosis [9]. The neurotropic nature of EBV and HCMV makes them particularly dangerous, contributing to conditions ranging from systemic infections to cancer [10].

In regions such as the western Brazilian Amazon, where the prevalence and mortality secondary to HIV/AIDS are significant, and on the higher end of the country, HHVs are believed to pose a greater risk to this population [11]. These individuals are particularly susceptible to severe HHV-related complications due to compromised immune systems, often exacerbated by the interruption of antiretroviral therapy.

Given the broad prevalence of HHVs and their significant impact on public health, particularly among immunocompromised populations, this study aims to retrospectively analyze the effect of HHVs on patients with neurological symptoms, focusing on the Herpesviridae family’s contributions to CNS infections. We detailed the prevalence and clinical manifestations of these infections, aiming to enhance the understanding of the epidemiological scenario and exploring aspects of pathogenesis to further improve patient management.

## 2. Materials and Methods

### 2.1. Study Design and Site

This is a retrospective cohort study carried out between January 2015 and December 2022. The study was conducted at the Fundação de Medicina Tropical Doutor Heitor Vieira Dourado (FMT-HVD), a reference tertiary care hospital for infectious diseases in Manaus, Brazil. The institution operates in the public health system in Brazil (SUS) with no charges for patients. The FMT-HVD receives patients with suspected neurological diseases and CSF samples for diagnosis from hospitals in Manaus and surrounding cities.

### 2.2. Ethical Considerations

The Ethics Review Board of FMT-HVD approved this study (CAAE:64565822.7.0000.0005) as per the guidelines and standards for regulating research on human subjects established in Resolution 466/12, of the National Health Council of the Brazilian Ministry of Health. A waiver of informed consent was obtained due to the retrospective nature of the study. Patient anonymity was preserved throughout data extraction and analysis.

### 2.3. Study Population and Sample Collection

The study cohort included patients with suspected CNS infections, particularly acute meningitis, encephalitis, and meningoencephalitis. Lumbar punctures were performed to collect CSF samples from all patients upon admission. The CSF was analyzed for herpesvirus infections and subjected to routine assessments, including total and differential cell counts, protein, glucose, lactate levels via spectroscopy, and microbiological testing for bacteria and fungi (smear, culture, latex), in addition to molecular diagnostics using qPCR. A subset of these patients tested positive for HIV.

Patients with herpesvirus detected in their CSF were identified retrospectively from the laboratory database. Exclusion criteria included patients with suspicions of other pathogens, those who had experienced trauma, or those with repeated CSF sampling.

### 2.4. Data Collection

We collected sociodemographic, clinical, and laboratory data. The categories included signs and symptoms, types of infection, therapeutic regimen, viral load, CD4+/CD8+ T lymphocyte counts, hospitalization period, laboratory tests, and CSF profile. All data were obtained via the electronic medical record. The information was managed in a database using the software RedCap© (version 12.2.10 Vanderbilt University, Nashville, TN, USA, 2002).

### 2.5. Case Definition

Viral meningitis is a neurological disease with inflamatory symptoms affecting the meninges, the protective membranes covering the brain and spinal cord (dura mater, arachnoide, and pia mater). Manifesting acutely or chronically, it presents distinctive indicators such as nuchal rigidity (positive Brudzinski’s or Kering’s sign), fever, vomiting, photosensibility, and changes in CSF. Diagnosed individuals commonly exhibit CSF leukocyte counts of 80 to 100 cells/μL, known as pleocytosis [12].

In contrast, viral encephalitis is a severe neurological disease-causing inflammation in the brain parenchyma, resulting in focal neurological, decreased cognitive and alert, and motor deficits, along with higher mortality and long-term sequelae. Suspicions of viral etiology arise with symptoms such as severe headache, decreased alertness, seizures, and behavioral changes, coupled with a fever of ≥38 °C. CSF specimens from those with encephalitis typically show lymphocytic pleocytosis, a slight increase in proteinorraquia, and normal glucose levels [13].

### 2.6. Molecular Diagnosis

Nucleic acids were extracted from 200 μL of cerebrospinal fluid (CSF) using the ReliaPrep™ Viral TNA MiniPrep system (Promega, Madison, WI, USA). Quantitative PCR (qPCR) reactions were prepared using the GoTaq^®^ Probe 1-Step RT-qPCR System (Promega, Madison, WI, USA). The reaction mix included 10 μL of Master Mix, 5.5 μL of nuclease-free water, 1.5 μL of an Assay-by-Design primer and probe set, and 3 μL of DNA, resulting in a total volume of 20 μL. Specific genes, including herpes simplex virus types 1 and 2 (HSV-1/2), Epstein–Barr virus (EBV), Varicella-zoster virus (VZV), and Human Cytomegalovirus (HCMV), were amplified using singleplex PCR [14,15].

Primer and probe sets were selected, and a synthetic positive external control, encompassing target regions for HSV-1, HSV-2, CMV, VZV, and EBV, was custom-designed using pGBLOCK by IDT DNA Technology (Coralville, IA, USA). The thermocycler conditions for the real-time PCR system were set at 45 °C for 15 min, 95 °C for 2 min, followed by 40 cycles of 95 °C for 15 s and 60 °C for 1 min. Each qPCR reaction included CSF samples, a positive external control, a negative control (nuclease-free water), and an internal control involving β-actin amplification to confirm the presence of nucleic acids.

### 2.7. Data Analysis

A Shapiro-Wilk normality test was conducted on various variables to assess the data distribution. Statistical summary tables were prepared for the variables of interest, grouped by type of neurological disease (meningitis, encephalitis, meningoencephalitis). The tables present data such as medians, quartiles, frequencies, and percentages for continuous and categorical variables. In the general table, chi-square tests (for parametric categorical variables) and ANOVA (for parametric continuous variables) were performed to investigate associations between the variables. In the HIV-related table, chi-square tests (for parametric categorical variables) and Kruskal-Wallis tests (for non-parametric continuous variables) were employed. In the table with laboratory data, Kruskal-Wallis tests (for non-parametric continuous variables) and ANOVA (for parametric continuous variables) were used. The results are highlighted in the general table by hospital discharges and symptoms, while in the HIV table, results such as HIV stage, hospital discharges, and symptoms are observed.

## 3. Results

### 3.1. Patient Characteristics and Clinical Diagnosis

This study analyzed 895 samples from patients with suspected CNS viral infections at the FMT-HVD. After excluding samples without data, duplicates, negative, and tested for another pathogen in CSF (n = 828), 7.5% (67/895) tested positive for HHVs. A total of 34 (50.7%) individuals were male, ranging from 1 month to 91 years old. Among the 67 patients, HCMV occurred in 26.8% (n = 18), EBV in 25.3% (n = 17), VZV in 20.8% (n = 14), HSV-2 in 8.9% (n = 6), and HSV-1 in 7.4% (n = 5). Coinfections occurred in 10.4% (n = 7), including VZV + EBV (n = 4), HCMV + EBV (n = 2), and HCMV + VZV (n = 1) (Figure 1). Clinical diagnoses based on symptoms and tests included encephalitis (46.2%, n = 31), meningitis (31.3%, n = 21), meningoencephalitis (11.9%, n = 8), and 10.4% (n = 7) without a specific neurological diagnosis. Notably, 58.2% (n = 39) were PLWHA. All deaths occurred in PLWHA (20.9%, n = 14) (Table 1).

### 3.2. Neurological Manifestations

Typical acute CNS infection symptoms were present in 89.5% (n = 60), primarily fever (77%, n = 46), vomiting (50%, n = 30), headache (67%, n = 40), and neck stiffness (32%, n = 19). A total of 62% (n = 37) experienced reduced alertness, 27% (n = 16) muscle weakness, and 15% (n = 9) seizures (Table 1). Comparative analysis showed encephalitis patients had significantly more consciousness alterations (*p* < 0.001) than those with meningitis or meningoencephalitis. The Chi-Square test showed a lower headache distribution in encephalitis patients (*p* = 0.036). Encephalitis and meningoencephalitis patients also showed greater alertness reduction (*p* < 0.001) and a higher prevalence of neck stiffness (*p* < 0.001) compared to meningitis patients. Seizure was more prevalent in encephalitis patients (*p* = 0.007) (Table 1).

### 3.3. PLWHA and Herpesviruses

Among 36 PLWHA, 52.7% (n = 19) were diagnosed with encephalitis. Compared to encephalitis patients, those with meningitis and meningoencephalitis presented more episodes of fever (*p* = 0.040), headache (*p* = 0.032), and neck stiffness (*p* < 0.001). Encephalitis and meningoencephalitis patients also showed greater alertness reduction (*p* < 0.001), with more frequent seizures noticed in encephalitis patients (*p* = 0.040). Most encephalitis patients (66%, n = 24) presented AIDS, and meningoencephalitis patients often had an unknown clinical stage of HIV (*p* = 0.037). Mortality reached 39% (n = 14), with meningitis patients showing a higher hospital discharge rate (*p* = 0.022), though mortality did not significantly differ between groups (Table 2).

### 3.4. CSF Sample Profile

CSF analysis showed no bacterial growth or latex agglutination test positivity. 56.5% (n = 39) were colorless, and 27.9% (n = 19) were xanthochromic. Most samples were clear (69.1%, n = 47), with 94.2% (n = 65) not forming a clot. Protein analysis found 27.9% (n = 19) of samples globulin-free. The CSF profile showed moderate alteration, with a median cell count of 151 cells/mm^3^, lymphocytes (8–906), median protein of 108 mg/dL, and glucose within the reference range (54 mg/dL). No statistical significance was found in group comparisons (Table 3).

### 3.5. Comparison between PLWHA and Non-HIV

A total of 39 (58.2%) individuals were PLWHA. Encephalitis was more prevalent among PLWHA in comparison to the HIV-negative group, and EBV was more common among PLHWA (*p* = 0.04). PLWHA had longer hospital stays (*p* < 0.05) in comparison to non-HIV individuals (Table 4). Most CNS herpesvirus patients were PLWHA, underscoring the Herpesviridae viruses’ opportunistic nature. Among 39 PLWHA, 36% (n = 14) had other opportunistic diseases, including oral candidiasis (n = 5), syphilis (n = 1), neurosyphilis (n = 1), neurocryptococcosis (n = 2), tuberculosis (n = 1), toxoplasmosis (n = 5), pneumocystosis (n = 1), and histoplasmosis (n = 1).

## 4. Discussion

In our study conducted at a Manaus infectious disease reference center, human herpesviruses were linked with 7.5% of CNS infections over eight years. Notably, 51.6% of these infections were cases of encephalitis, with HCMV accounting for 28.3% and EBV for 26.6%. This diverges from prior research highlighting HSV-1 and VZV as primary viral agents of encephalitis [16,17]. Although HCMV and EBV reactivation is often triggered by HIV-associated immunosuppression, they remain significant neurological pathogens. These viruses are well-documented causes of severe CNS infections such as encephalitis and meningitis [4], and their reactivation in HIV patients underscores the need for vigilant monitoring and effective treatment. Furthermore, the prominence of EBV and HCMV in this study may have been due to the limited antiviral options for HCMV and the lack of effective treatments for EBV, in contrast to the efficacy of acyclovir against HSV.

Meningitis and encephalitis differ in their inflammation of the meninges and brain parenchyma, respectively. Meningitis presents with severe headache and neck stiffness, whereas encephalitis may cause motor deficits, seizures, and behavioral changes [18,19]. These symptoms are often exacerbated by low CD4+ T-cell counts, chronic diseases, or treatment-induced lack of adherence [20]. In this study, individuals with encephalitis frequently led to altered consciousness and seizures. The reduced headache incidence in encephalitis cases supports this distinction.

CNS infections are often early AIDS indicators due to their inflammatory nature [21]. In this cohort, PLWHA presented meningoencephalitis and encephalitis. We noted a 20% mortality rate among these patients, underscoring the high mortality linked to HIV infection, prolonged hospitalization, and pre-existing conditions. This aligns with a Chinese study associating mortality with poor antiretroviral adherence and discontinuation of clinical follow-up [22].

The blood-brain barrier (BBB) and blood-cerebrospinal fluid barrier (BCSFB) play a critical role in protecting the CNS from various pathogens. These barriers function through a series of continuously active mechanisms, specifically expressed by endothelial cells and brain capillaries. These mechanisms facilitate the entry of essential nutrients into the CSF while excluding potentially harmful blood molecules, thereby maintaining the environment necessary for neural transmission. Damage to the BBB and BCSFB can lead to leukocyte infiltration into the CSF, triggering a CNS inflammatory response, as indicated by changes in CSF parameters. The presence of blood-derived proteins in the CSF suggests barrier dysfunction, which correlates with CNS damage [23]. Our findings of pleocytosis and increased CSF protein levels in herpesvirus infections suggest barrier compromise, though definitive damage cannot be conclusively determined.

In healthy adults, the CSF may contain up to 5 leukocytes per mm^3^, whereas cases of viral meningitis often show fewer than 100 cells/mm^3^ [19]. The variability in cell counts and the lack of direct correlation between viral presence and specific neurological diseases in our study indicated that herpesviruses are related to cases of meningitis, encephalitis, or meningoencephalitis. Considering this, we recommend regular CSF biochemical analyses and molecular diagnosis (qPCR) for immunocompromised patients with neurological symptoms.

Although our results indicate a significant association between HHVs, particularly HCMV and EBV, and CNS infections, it is important to note that the detection of these viruses in CSF samples does not necessarily prove a direct causal role in the observed neurological symptoms. The reactivation of latent viruses, especially in immunocompromised individuals such as PLWHA, may explain the presence of these viruses without implying direct causality. Therefore, conclusions about the causal role of HHVs should be interpreted with caution. We recommend that future research focus on establishing a clearer causal relationship between the presence of HHVs and neurological symptoms, considering other factors that may contribute to the development of these CNS infections.

Data collection limitations, including incomplete medical records and challenges in obtaining external hospital data, restricted our study. The prevalence of PLWHA, particularly those on prophylactic antiviral therapy, may have obscured positive diagnoses. Therefore, the antiviral can reduce the viral load of HHVs in CSF and other body fluids, which can lead to false-negative results in diagnostic tests, especially in PCR-based assays that rely on the detection of viral DNA. Consequently, the sensitivity of diagnostic methods may have been compromised, reducing the apparent prevalence of herpes simplex virus types 1 and 2 infections.

## 5. Conclusions

Our eight-year investigation at a Manaus infectious disease center revealed a 7.5% CNS infection rate due to HHV. Encephalitis and meningoencephalitis, especially prevalent among PLWHA, pose substantial diagnostic and therapeutic challenges due to symptom complexity and immunosuppression effects. Our CSF findings, including pleocytosis and protein increase, underscore the importance of thorough CSF analysis in diagnosing CNS infections. This study enhances our understanding of viral CNS infections amidst high HIV prevalence, stressing the need for patient-tailored diagnostic and treatment approaches. Ongoing research and clinical practice advancements are vital for improving outcomes in CNS infection patients, particularly the immunocompromised.

## Figures and Tables

**Figure 1 viruses-16-01437-f001:**
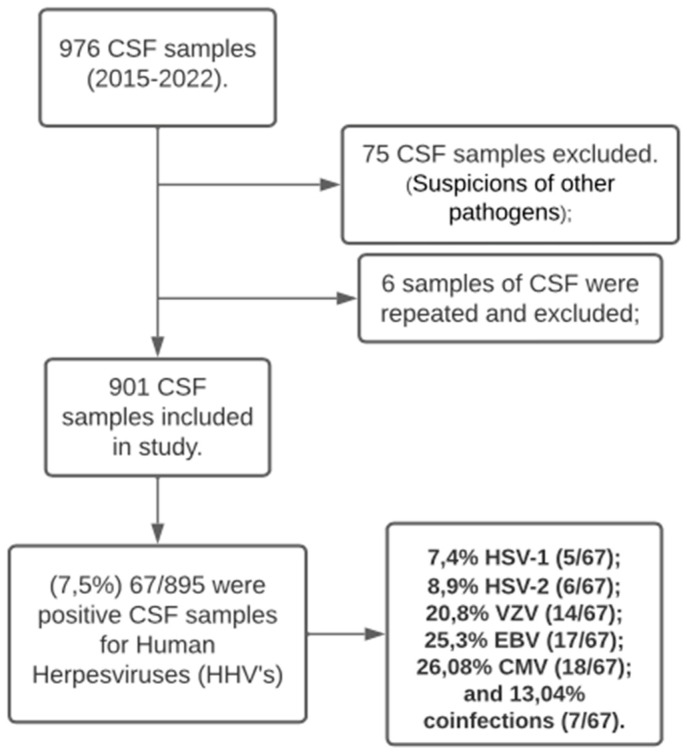
Flowchart of the study.

**Table 1 viruses-16-01437-t001:** Demographic, etiological, and neurological manifestations in herpesvirus neurological patients, Manaus, Brazil (2015–2022).

Demografic Caracteristic	Meningitis (N = 21)	Encephalitis (N = 31)	Meningoencephalitis (N = 8)	*p* Value
Sex (masc:fem)	14:7	13:18	3:5	0.2
Age [Md]	34.4 [7–91]	36.3 [<1–77]	43.2 [23–65]	0.4
Etiological Agent				
HSV-1 ^1^	2 (9.5%)	1 (3.2%)	1 (12.5%)	0.8
HSV-2 ^2^	1 (4.7%)	4 (13%)	1 (12.5%)	
VZV ^3^	6 (28.5%)	6 (19.3%)	0	
EBV ^4^	4 (19%)	9 (29%)	3 (37.5%)	
HCMV ^5^	7 (33.3%)	8 (25.8%)	2 (25%)	
Coinfections	1 (4.7%)	3 (9.6%)	1 (12.5%)	
Symptoms				
Fever	17 (81%)	21 (68%)	8 (100%)	0.13
Headache	17 (81%)	16 (52%)	7 (88%)	0.036
Vomit	10 (48%)	13 (42%)	7 (88%)	0.069
Neck stiffness	12 (57.1%)	1 (3.2%)	6 (75%)	<0.001
Decreased alertness	2 (9.5%)	28 (90.3%)	7 (87.5%)	<0.001
Seizure	0	9 (29%)	0	0.007
Muscle weakness	5 (24%)	10 (32.2%)	1 (12.5%)	0.5
HIV ^6^				
Yes	11 (52.3%)	19 (61.2%)	6 (75%)	0.6
No	10 (47.6%)	12 (38.7%)	2 (25%)	
Outcome				
Discharge	19 (90.4%)	14 (67.7%)	3 (43%)	0.021
Died	2 (9.5%)	10 (32.2%)	2 (25%)	0.11

^1^—HSV-1, *Herpes simplex virus* 1; ^2^—HSV-2, *Herpes simplex virus* 2; ^3^—VZV, *Varicella-zoster* virus; ^4^—EBV, *Epstein–Barr* virus; ^5^—HCMV, Human *Cytomegalovirus*; ^6^—HIV, Human immunodeficiency virus.

**Table 2 viruses-16-01437-t002:** Neurological symptoms, stage of AIDS, and outcome in neurological patients with herpesvirus in Manaus, Brazil, between 2015 and 2022.

Symptoms of PLWHA ^1^	Meningitis N = 11 (%)	Encephalitis N = 19 (%)	Meningoencephalitis N = 6 (%)	*p* Value
Fever	10 (91%)	11 (58%)	6 (100%)	0.040
Vomit	5 (45%)	8 (42%)	5 (83%)	0.2
Headache	11 (100%)	11 (58%)	5 (83%)	0.032
Neck stifness	5 (45.4%)	0	4 (66.6%)	<0.001
Decreased alertness	2 (18%)	19 (100%)	5 (83%)	<0.001
Seizure	0	6 (32%)	0	0.040
Muscle weakness	4 (36%)	7 (37%)	0	0.2
Stage of AIDS ^2^				
Stage 1 * (n = 1)	0	0	0	-
Stage 2 ** (n = 6)	2 (18%)	3 (16%)	1 (17%)	<0.9
Stage 3 *** (n = 24)	7 (64%)	15 (79%)	2 (33%)	0.11
Unknow stage **** (n = 6)	2 (18%)	1 (5.3%)	3 (50%)	0.037
Outcome				
Died	2 (18%)	10 (53%)	2 (33%)	0.2
Discharge	9 (82%)	6 (32%)	2 (33%)	0.022

^1^—PLWHA, people living with HIV/AIDS: and ^2^—AIDS, Acquired Immune Deficiency Syndrome. *: Stage 1 > 500 TCD4+ cels/mm^3^, **: Stage 2 > 200 TCD4+ cels/mm^3^, ***: Stage 3 < 200 TCD4+ cels/mm^3^, and ****: Unknown Stage, No TCD4+ cell count information.

**Table 3 viruses-16-01437-t003:** Comparison of the cerebrospinal fluid profile between the groups of patients who were diagnosed with meningitis, encephalitis, and meningoencephalitis in Manaus, Brazil, between January 2015 and December 2022.

Biochemical Parameters	Meningitis N = 21	Escephalitis N = 31	Meningoencephalitis N = 8	*p* Value
Cytometry (cell/mm^3^)	85 (5–320)	182 (31–480)	259 (106–473)	<0.9
Glucose (mg/dL)	64 (45–75)	54 (45–68)	77 (42–105)	0.4
Proteins (mg/dL)	121 (59–161)	139 (66–206)	93 (68–133)	0.6
Lactate (mmol/L)	4 (3–24)	20 (4–34)	32 (10–42)	0.053

**Table 4 viruses-16-01437-t004:** Neurological diseases, viral agents, clinical features, and outcome in PLWH and not HIV in Manaus, Brazil, between 2015 and 2022.

Variables (Total) Neurological Disease (N = 67)	PLWHA ^1^ (N = 39)	Non-HIV/AIDS ^2^ (N = 28)	*p* Value
Meningitis (n = 21)	10 (25.6%)	11(39.2%)	0.35
Encephalitis (n = 31)	19 (48.7%)	12 (42.8%)	0.82
Meningoencephalitis (n = 8)	6 (15.3%)	2 (7.14%)	0.45
No information of diagnosis (n = 7)	3 (7.7%)	4 (14.2%)	0.43
Viral agent			
HCMV ^3^ (n = 18)	10 (25.6%)	8 (28.55)	0.99
EBV ^4^ (n = 17)	14 (35.8%)	3 (10.7%)	0.04
VZV ^5^ (n = 14)	7 (18%)	7 (25%)	0.69
HSV-2 ^6^ (n = 6)	0	6 (21.4%)	-
HSV-1 ^7^ (n = 5)	1 (2.5%)	4 (14.2%)	0.15
Symptons			
Fever (n = 50)	28 (71.8%)	22 (78.5%)	0.73
Vomit (n = 32)	19 (48.7%)	13 (46.4%)	0.94
Headache (n = 44)	29 (74.3%)	15 (53.5%)	0.13
Neckstifness (n = 19)	9 (23%)	10 (35.7%)	0.39
Decreased alertness (n = 39)	26 (66.6%)	13 (46.4%)	0.15
Seizure (n = 9)	6 (15.3%)	3 (10.7%)	0.72
Muscle weakness (n = 18)	12 (30.7%)	6 (21.4%)	0.56
Outcome and hospitalization			
Discharge	25 (64.1%)	28 (100%)	-
Died	14 (35.89%)	0	
Hospital care (days)	1–245 (39.6)	3–35 (12.4)	<0.05

^1^—PLWHA, people living with HIV/AIDS; ^2^—non-HIV/AIDS, people not living with HIV/AIDS; ^3^—HCMV, Human *Cytomegalovirus*; ^4^—EBV, *Epstein–Barr virus*; ^5^—VZV, *Varicella zoster virus*; ^6^—HSV-2, *Herpes simples virus* type 2: and ^7^—HSV-1, *Herpes simples virus* type 1.

## Data Availability

Data will be made available on request.

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
