# Peer review of "Unveiling the Impact of Human Herpesviruses-Associated on CNS Infections: An Observational Study"

_viruses, 2024, doi:10.3390/v16091437_

Round 1

Reviewer 1 Report

Comments and Suggestions for Authors

The role of HHV in neurological disease is of interest but has been addressed many times using clinical samples. This study fails to move us forward because it tries to do much. Issues to be considered include

1. the timing of sample collection relative to symptom onset...and hence relative to the presence of viral DNA

2. co-morbidities such as TB in the PLWH

I would recommend a re-write in a shorter form with the Tables  designed to tell your reader about the manifestations seen in cases positive for each HHV. In a single Table this could show whether each HHV was more common in PLWH. Anecdotal data re ART would then be interesting.

Comments on the Quality of English Language

The manuscript needs checking for English grammar. With care, the length can be reduced by 50% with no loss of content.

Author Response

To:  Viruses

Chief Editor

New title: Unveiling the Impact of Human Herpesviruses-associated on CNS Infections: An Observational Study

We thank you and the reviewers for the insightful comments and suggestions, which have improved our manuscript considerably. Below, you will find a point-by-point response to the reviewer's comments. The modifications have been highlighted in yellow in the revised manuscript.

We believe that the reviewers' very relevant comments and suggestions have improved the quality and readability of the manuscript. 

We are extremely honored and proud to be considered for publication in the Viruses, which recognizes its impact on critical care and broad exposure to a large audience.

Sincerely yours,

Fernando Almeida-Val, PhD.

REVIEWER 1

Comments and Suggestions for Authors: The role of HHV in neurological disease is of interest but has been addressed many times using clinical samples. This study fails to move us forward because it tries to do much. Issues to be considered include.

The timing of sample collection relative to symptom onset...and hence relative to the presence of viral DNA

Authors' response: We thank the reviewer for the comment and believe this data could add interesting information in this regard. Nonetheless, due to the retrospective nature of the study, much information could not be retrieved from the medical records, particularly the date of symptom onset, which was not systematically collected at the time of the diagnosis. Therefore, the requested data was not available and will not be inserted in the manuscript.

REVIEWER 2

Comments and Suggestions for Authors: Barrionuevo et al tested by PCR for herpesviruses CSF of 945 patients diagnosed with various CNS diseases like meningitis, encephalitis. 7.5% of the samples proved to be positives, 5 herpesviruses were detected. The patients were partly HIV infected. The applied methods are appropriate, although I could not understand what was the main messeage of the paper. Herpesviruses could be reactivated, the authors do not explain whether what kind of role they suspect the herpesviruses played in these diseases? Effect of HIV infection on the dianostic results should also be discussed in more detail.

Authors' response: We thank the reviewer for their consideration, we have improved the article's message in Pag 9.

“Although HCMV and EBV reactivation is often triggered by HIV-associated immunosuppression, they remain significant neurological pathogens. These viruses are well-documented causes of severe CNS infections such as encephalitis and meningitis [4], and their reactivation in HIV patients underscores the need for vigilant monitoring and effective treatment.”

The English of the paper is good, although there are several so evident spelling mistakes, which indicate that authors did not read the text even once after writing it.

Authors' response: Thanks for the comment. We improved the quality of English.

Abstract: No abbreviation in abstract: PLWHA. What is it? Give in full in the text (also BBB). In line 23: people living with HIV/AIDS.

Authors' response: We include the definitions of the acronyms.

Results: The findings revealed that 7.5% of the analyzed samples tested positive for HHVs, with Human Cytomegalovirus (HCMV) and Epstein-Barr Virus (EBV) being the most prevalent. A significant association was found between HHVs and neurological diseases such as encephalitis and meningitis, especially among people living with HIV/AIDS (PLWHA), highlighting the opportunistic nature of these viruses.

Introduction: Paragraph 41-47. Prevalence, latency/reactivation+vaccination problems. Not related things one after the other. What is this paragraph about?. line 48 - All HHVs infect the CNS?

Authors' response: We include this information in introduction:

These five HHVs can infect the CNS, leading to severe conditions such as encephalitis and meningitis [8], though they do not always result in neurological syndromes. They also contribute to other serious health issues, including autoimmune diseases and neurodegenerative disorders like Alzheimer’s, Parkinson’s, and Multiple Sclerosis [9].

Meth: Line 82 - All patients were HHV +? Detected by which method?

Authors' response: We include this information in methods: 67 (7.5%) were HHV positive. Diagnosis was performed by molecular diagnostics (PCR). We have added the following paragraph in the methods section:

2.6. Molecular diagnosis

Nucleic acids were extracted from 200μL of cerebrospinal fluid (CSF) using the Re-liaPrep™ Viral TNA MiniPrep system (Promega, WI, USA). Quantitative PCR (qPCR) reactions were prepared using the GoTaq® Probe 1-Step RT-qPCR System (Promega, WI, USA). The reaction mix included 10μL of Master Mix, 5.5μL of nuclease-free water, 1.5μL of an Assay-by-Design primer and probe set, and 3μL of DNA, resulting in a total volume of 20μL. Specific genes, including herpes simplex virus types 1 and 2 (HSV-1/2), Epstein–Barr virus (EBV), Varicella-zoster virus (VZV), and Human Cytomegalovirus (HCMV), were amplified using singleplex PCR [14,15].

Primer and probe sets were selected and a synthetic positive external control, en-compassing target regions for HSV-1, HSV-2, CMV, VZV, and EBV, was custom-designed using pGBLOCK by IDT DNA Technology (IA, USA). The thermocycler conditions for the real-time PCR system were set at 45°C for 15 minutes, 95°C for 2 minutes, followed by 40 cycles of 95°C for 15 seconds and 60°C for 1 minute. Each qPCR reaction included CSF samples, a positive external control, a negative control (nuclease-free water), and an internal control involving β-actin amplification to confirm the presence of nucleic acids.

What proportion of HIV infected persons carried HHVs in the CNS?

Authors' response: 36 (60%). This was added to the manuscript.

121-122 - Human cytomegalovirus. EBV, VZV, HSV1 and 2 are specific viruses, but cytomegalovirus is a group of viruses with lot of members. It should be mentioned as HHV-5, or human cytomegalovirus.

Authors' response: Thanks for the observation. We have corrected it to “human cytomegalovirus (HCMV)”.

Lines 196 -198. Could the authors state that herpesviruses played  causative roles in these meningitis, encephalitis cases? Or the HIV infection, with its immunsuppressive virus stands behind these illnesses, symtoms? if the herpesviruses are only there as a result of reactivations from latency, are they important in this respect? All illness, immunosuppressions, operations, constant stress are able to provoke such reactivations. HIV infections are the best among these impacts.

Authors' response: Thank you for the observation. We cannot say for certain that, as the reviewer mentioned. Despite the significant detection of herpesviruses in CSF samples, this does not prove that these viruses are the direct cause of the observed neurological symptoms. Herpesviruses can remain latent in the body and be reactivated under conditions of immunosuppression, such as in patients with HIV/AIDS. The presence of viruses in CSF may be a result of this reactivation and not necessarily the primary cause of the neurological symptoms. And the detection of a virus may not be directly related to the cause of the disease; this viral presence may be secondary to another underlying condition.

Therefore, we have included the following information in the discussion:

Although our results indicate a significant association between HHVs, particularly HCMV and EBV, and CNS infections, it is important to note that the detection of these viruses in CSF samples does not necessarily prove a direct causal role in the observed neurological symptoms. The reactivation of latent viruses, especially in immunocompromised individuals such as PLWHA, may explain the presence of these viruses without implying direct causality. Therefore, conclusions about the causal role of HHVs should be interpreted with caution. We recommend that future research focus on establishing a clearer causal relationship between the presence of HHVs and neurological symptoms, considering other factors that may contribute to the development of these CNS infections.

Table 4. It seems to me, that EBV is there mostly because of reactivation caused HIV and as such, this agent can not be considered as a neurological pathogen. While HSV-2 is there only in HIV-free patients. How the authors could explain that?

Authors' response: Although EBV reactivation is triggered by HIV-associated immunosuppression, this does not exclude EBV from being considered a neurological pathogen. Its ability to cause serious CNS infections such as encephalitis and meningitis is well documented, and reactivation in patients with HIV only increases the importance of monitoring and treating these infections effectively.

HIV patients who present with neurological manifestations are often given prior antiviral therapy with acyclovir, which can reduce the viral load of HSV1 and 2 in cerebrospinal fluid (CSF) and other body fluids. This can lead to false-negative results in diagnostic tests, especially in PCR-based assays that rely on detection of viral DNA. Consequently, the sensitivity of diagnostic methods may be compromised, leading to a reduction in the prevalence of herpes simplex type 1 and 2 infections. However, immunocompetent patients rarely use the antiviral acyclovir beforehand, so the prevalence of this virus may have been higher in this group of patients.

213-215 – The authors should not forget, that these results do not prove, that herpesviruses played causative roles in the diagnosed CNS symptoms. The detection of a herpesvirus ( which can be reactivated from latent state) from a tissue, illness do not prove causative roles of the agents. The authors should state this somewhere in the discussion, and in many rows phrases which consider herpesvirus as causative agents should be rewritten.

Authors' response: We appreciate the reviewer’s comments in this regard. To further clarify this issue, we have included the following information in the discussion:

In healthy adults, the CSF may contain up to 5 leukocytes per mm³, whereas cases of viral meningitis often show fewer than 100 cells/mm³ [19]. The variability in cell counts and the lack of direct correlation between viral presence and specific neurological diseases in our study indicated that herpesviruses are related to cases of meningitis, encephalitis, or meningoencephalitis. Considering this, we recommend regular CSF biochemical analyses and molecular diagnosis (qPCR) for immunocompromised patients with neu-rological symptoms.

Although our results indicate a significant association between HHVs, particularly HCMV and EBV, and CNS infections, it is important to note that the detection of these viruses in CSF samples does not necessarily prove a direct causal role in the observed neurological symptoms. The reactivation of latent viruses, especially in immunocom-promised individuals such as PLWHA, may explain the presence of these viruses without implying direct causality. Therefore, conclusions about the causal role of HHVs should be interpreted with caution. We recommend that future research focus on establishing a clearer causal relationship between the presence of HHVs and neurological symptoms, considering other factors that may contribute to the development of these CNS infections.

Data collection limitations, including incomplete medical records and challenges in obtaining external hospital data, restricted our study. The prevalence of PLWHA, particularly those on prophylactic antiviral therapy, may have obscured positive diagnoses. Therefore, the antiviral can reduce the viral load of HHVs in CSF and other body fluids, which can lead to false-negative results in diagnostic tests, especially in PCR-based assays that rely on the detection of viral DNA. Consequently, the sensitivity of diagnostic methods may have been compromised, reducing the apparent prevalence of herpes simplex virus types 1 and 2 infections.

Discussion

223-24 – „treatment non-adherence”, what is it? line 233 – BBB. abbreviation, was it given full somewhere?

Authors' response: We have included definitions for all acronyms:

Blood–brain barrier (BBB)

line 241 – if not, what is the message of this paper?

Authors' response: We have included the following information in the discussion:

“The cell counts variability and lack of direct correlation between viral presence and specific neurological diseases in our study may indicate that herpesviruses may have been the agents related to the causes of meningitis, encephalitis, or meningoencephalitis. Therefore, conclusions about the causal role of herpesviruses should be interpreted with caution.”

Comments on the Quality of English Language:

line 23. abbreviations should be avoided in the Abstract chapter. PLWHA? line 196??? Give the full name at the first appearance in the text. Is it given somewhere?

lines 37, 39 – Gammaherpesvirinae

line 100 (2002).

line 103 – Coverging?

104 – pr

105 – Sush/such

103-105 had anyone read these rows?

117- TNA????

line 122 – (CMV), (S, a space is missing

Figure 1 – Tested for other

Table 2. stiffness

Table 4: diseases

Authors' response: We have included definitions for all acronyms, improved the writing and highlighted them in yellow in the article.

Reviewer 2 Report

Comments and Suggestions for Authors

Barrionuevo et al tested by PCR for herpesviruses CSF of 945 patients diagnosed with various CNS diseases like meningitis, encephalitis. 7.5% of the samples proved to be positives, 5 herpesviruses were detected.The patients were partly HIV infected. The applied methods are appropriate, although I could not understand what was the main messeage of the paper. Herpesviruses could be reactivated, the authors do not explain whether what kind of role they suspect the herpesviruses played in these diseases? Effect of HIV infection on the dianostic results should also be discussed in more detail.

The English of the paper is good, although there are several so evident spelling mistakes, which indicate that authors did not read the text even once after writing it.

Abstract.

no abbreviation in abstract. PLWHA. What is it? Give in full in the text (also BBB).

Introduction:

paragraph 41-47. Prevalence, latency/reactivation+vaccination problems. Not related things one after the other. What is this paragraph about?

line 48 - All HHVs infect the CNS?

Mat.meth.

line 82 - All patients were HHV +? Detected by which method?

line 89 - How HIV and HHV infections coincided? What proportion of HIV infected persons carried HHVs in the CNS?

95 -100 - How were these data groups related to HHV infections?

104 -107 - The authors should indicate, that these are not specific symptoms.

121-122 - Human cytomegalovirus. EBV, VZV, HSV1 and 2 are specific viruses, but cytomegalovirus is a group of viruses with lot of members. It should be mentioned as HHV-5, or human cytomegalovirus.

Figure 1 - 75 samples were excluded from the study, because other microorganisms were detected from the CSF. What were these agents? Their proportion? And if herpeviruses were also were there?

lines 196 -198. Could the authors state that herpesviruses played  causative roles in these meningitis, encephalitis cases? Or the HIV infection, with its immunsuppressive virus stands behind these illnesses, symtoms? if the herpesviruses are only there as a result of reactivations from latency, are they important in this respect? All illness, immunosuppressions, operations, constant stress are able to provoke such reactivations. HIV infections are the best among these impacts.

Table 4. It seems to me, that EBV is there mostly because of reactivation caused HIV and as such, this agent can not be considered as a neurological pathogen. While HSV-2 is there only in HIV-free patients. How the authors could explain that?

213-215 – The authors should not forget, that these results do not prove, that herpesviruses played causative roles in the diagnosed CNS symptoms. The detection of a herpesvirus ( which can be reactivated from latent state) from a tissue, illness do not prove causative roles of the agents. The authors should state this somewhere in the discussion, and in many rows phrases which consider herpesvirus as causative agents should be rewritten.

row 215 – really caused? How do you know? When HIV infections are necessary to detect EBV in most cases, how could You consider EBV as causative agent of the CNS illnesses?

Discussion

223-24 – „treatment non-adherence”, what is it?

line 233 – BBB. abbreviation, was it given full somewhere?

line 241 – if not, what is the meseage of this paper?

line 244 – How many HIV infected persons got antiviral therapy, and how this could influence Your HHV data?

Comments on the Quality of English Language

Spelling:

line 23. abbreviations should be avoided in the Abstract chapter. PLWHA? line 196??? Give the full name at the first appearance in the text. Is it given somewhere?

lines 37, 39 – Gammaherpesvirinae

line 100 (2002).

line 103 – Coverging?

104 – pr

105 – Sush/such

103-105 had anyone read these rows?

117- TNA????

line 122 – (CMV), (S, a space is missing

Figure 1 – Tested for other …..

Table 2. stiffness

Table 4: diseases

Author Response

(The authors gave the same response as above.)

Round 2

Reviewer 1 Report

Comments and Suggestions for Authors

This manuscript has been improved but problems remain

Table 1 has spelling errors. It also has an inadequate statistical analysis of the links with viral DNA….it should be line by line.

Percentages can be shown as whole numbers and decimal points should not be commas.

HIV can be a single line…and the data would be improved if a second line presented the patients on ART.

Was the overall column included in the stats?….perhaps remove the overall column to improve clarity.

Table 2 mixes PLWHA and PLWH…and wrongly defines the CDC classifications as stages of AIDS.

Was the overall column included in the stats?….perhaps remove the overall column to improve clarity.

Check the spelling.

Why highlight a difference in the “unknow stage” group….

Table 3 appears to have large differences between groups. The stats are not defined.

Table 4 …see comments above.

Comments on the Quality of English Language

The authors should check their work with more care. Spelling errors remain.

Author Response

  • Table 1 has spelling errors. It also has an inadequate statistical analysis of the links with viral DNA….it should be line by line.

Authors response: Thank you for your considerations. We have made corrections to table 1.

  • Percentages can be shown as whole numbers and decimal points should not be commas.

Authors response: Thank you for your considerations. We have corrected all percentage values, replacing commas with periods.

  • HIV can be a single line…and the data would be improved if a second line presented the patients on ART.

Authors response: We do not collect or enter information about ART.

  • Was the overall column included in the stats?….perhaps remove the overall column to improve clarity.

Authors response: Thank you for your considerations. We removed the overall column from all tables, so that interpretations are clearer, thus avoiding mistakes in interpreting the results.

Table 2 mixes PLWHA and PLWH…and wrongly defines the CDC classifications as stages of AIDS.

Authors response: Thank you for your observation. We have corrected the acronym from PLWH to PLWHA and added the definition of each stage of AIDS, according to the classification of the Center for Disease Control.

Stage 1 AIDS: the patient has a CD4+ T cell count > 500 cells/mm3;

Stage 2 AIDS: the patient has a CD4+ T cell count between 200 and 500 cells/mm3;

Stage 3 AIDS: the patient has a CD4+ T cell count > 200 cells/mm3;

Unknown Stage: when we are unable to retrieve information about the CD4+ T cell count.

  • Was the overall column included in the stats?….perhaps remove the overall column to improve clarity.

Authors response: Thank you for your observation. The column containing total values ​​was not included in the statistical analyses, but we accepted the suggestion to improve interpretations and avoid confusion.
